# UltraPrep is a scalable, cost-effective, bead-based method for purifying cell-free DNA

**Christopher K. Raymond[ID]**[1]*, **Fenella C. Raymond**[1], **Kay Hill**[2]

**1** Ripple Biosolutions, Seattle, Washington, United States of America, **2** PlasmaLab International, Everett, Washington, United States of America

* craymond@ripplebiosol.com

## Abstract

UltraPrep is an open-source, two-step method for purification of cell-free DNA that entails extraction of total DNA followed by size-selective enrichment of the smaller fragments that are characteristic of DNA originating from fragmentation between nucleosome. The advantages of the two related protocols that are described are that they can easily accommodate a wide range of sample input volumes, they rely on simple, magnetic bead-based technology, the yields of cfDNA are directly comparable to the most popular methods for cfDNA purification, and they dramatically reduce the cost of cfDNA isolation relative to currently available commercial methods. We provide a framework for physical and molecular quality analysis of purified cfDNA and demonstrate that the cfDNA generated by UltraPrep meets or exceeds the quality metrics of the most commonly used procedure. In addition, our method removes high molecular weight genomic DNA (hmwgDNA) that can interfere with downstream assay results, thereby addressing one of the primary concerns for preanalytical collection of blood samples.

## Introduction

Circulating cell-free DNA (cfDNA) has shown tremendous utility as an analyte in prenatal genetic analysis and in precision medicine approaches to diagnosed cancers. It holds promise to contribute to early detection of solid tumors [1,2]. This analyte has also shown potential for the rapid detection of infectious microbes [3]. Early detection tests that use cfDNA must be both highly sensitive and specific. Straightforward probability and statistics considerations indicate that this requires high input levels of cfDNA and subsequent genomic analysis that covers several thousand independent cfDNA "genome equivalents" [4,5]. In addition to individual patient testing, there is a largely unmet need for large, well-characterized, single-donor lots of normal human cfDNA that can be used for diagnostic test research, assay development, and routine proficiency qualification in clinical laboratory environments.

Plasmapheresis is a method that can be used to safely collect hundreds of milliliters of plasma from human subjects. Plasma is most often collected into vessels containing sodium citrate, a molecule that chelates divalent cations required by DNAse enzymes and thereby stabilizes extracellular DNA. Plasmapheresis samples contain significant quantities of cfDNA,

**Data Availability Statement:** All relevant data are within the manuscript and its Supporting Information files.

**Funding:** Funding for this study was provided jointly by Ripple Biosolutions and by PlasmaLab

International. These entities provided the resources necessary to conduct this study in the form of materials, supplies and salaries. All of the authors are employees and/or owners of these entities, as detailed below. CKR and FCR are co-founders and employees of Ripple Biosolutions, and both have an ownership stake in RB. KH is an owner and director of Plasma Lab International and has an ownership stake in PLI. The funder provided support in the form of salaries for authors, but did not have any additional role in the study design, data collection and analysis, decision to publish, or preparation of the manuscript. The specific roles of these authors are articulated in the 'author contributions' section.

**Competing interests:** All of the authors that contributed to this study are affiliated with either Ripple Biosolutions or PlasmaLab International as employees. In addition, the authors have an ownership stake in these entities. This affiliation does not alter our adherence to PLOS ONE policies on sharing data and materials. The open-source UltraPrep methods described in this manuscript are used to manufacture our commercial product, "UltraPure cfDNA".

and these can be used in the contexts described in the previous paragraph. At present, it is difficult to realize the potential of this cfDNA source owing to a lack of cost-effective methods for high volume cfDNA extraction and purification.

We were inspired by a recent publication promoting magnetic bead-based laboratory methods [6] to pursue an open-source approach to high-volume, reduced-cost purification of cfDNA. This proved to be a significant challenge. The most formidable obstacle was to achieve near-quantitative recovery of cfDNA fragments. Specifically, DNA binding to silica surfaces has been used as a purification method for decades [7,8], but a significant fraction of DNA is bound irreversibly [9,10]. Conditions for robust and reversible binding of cfDNA are reported here. In addition, preanalytical collection conditions often result in plasma that contains a mixture of hmwgDNA and cfDNA. By cfDNA, we mean a set of DNA fragments derived from cleavage between adjacent nucleosomes [11,12]. Since a single nucleosomal subunit is about 165 bp and cleavage between subunits can be incomplete, this results in a "ladder" of DNA fragments that are nucleosomal monomers, dimers, trimers, etc. [13]. This collection of "nucleosomal fragments" is thought to be generated by apoptosis that occurs among the cells in both normal and cancerous tissues. In contrast, hmwgDNA that is observed in some plasma samples is thought to be largely contributed by nucleated blood cells that burst. When present in quantities exceeding a few percent of the total cfDNA sample that is analyzed, excess hmwgDNA can result in underestimation of minor allele frequencies for somatic DNA variants, especially when using techniques such as quantitative digital PCR or amplicon-based DNA sequencing. In the case of cancer diagnostics, these minor allele frequencies translate directly into quantitative estimates of circulating tumor DNA burden, and these in turn may be used for treatment decisions in the clinic. Hence, there has been considerable effort invested in preanalytical collection methods that prevent cell lysis. Here we provide an alternative approach in which hmwgDNA can be removed from nucleosomal fragments by bead-based partitioning.

## Materials and methods

### Ethics statement

The ethics committees of Ripple Biosolutions and of Plasma Lab International reviewed and approved of the research presented here. Written consent was obtained from healthy donors prior to sample collection, processing and characterization.

### Materials

Plasma samples were collected at PlasmaLab International (Everett, WA) using automated plasmapheresis collection into sodium citrate containing collection vessels. The samples used for protocol development and described in S1 Table were stored at -20˚C for as long as 10 years. The $K_2$EDTA samples were derived from whole blood collected in standard vacutainer tubes that was spun immediately at 1500 g for 10 min. The top plasma fraction was transferred to a separate container and frozen at -20˚C prior to the cfDNA preparation. Lyophilized proteinase K (cat. P-480-5) was purchased from Gold Biotechnology (St. Louis, MO), guanidinium isothiocyanate (GITC) from Chem Impex (Wood Dale, IL), silica coated superparamagnetic beads (400–690 nm, cat. no. SIM-05-10H) from Spherotech (Lake Forest, IL), 1 M Tris pH 8.0 and 0.5 M EDTA from Quality Biologicals (Gaithersburg, MD), and isopropanol from Swan (Smyrna, TN). All other reagents for DNA purification were purchased from RPI Chemicals (Mount Prospect, IL). Magnetic rack bead separators were purchased from EBay (https://www.ebay.com/usr/pochekailov). We have not tested the performance of alternative 50 mL magnetic racks, but they appear to be widely available. Reagents for DNA

quantitation, DNA gel staining (Gel Green), and qPCR were from Biotium (Fremont, CA). Oligonucleotides were obtained from IDT (Coralville, IA). Molecular biology reagents for post-purification quality assessment were from New England Biolabs (Ipswich, MA). Fluorescent quantitation of DNA was measured on a Qubit instrument (ThermoFisher, Waltham, MA). DNA gels were run using the electrophoresis apparatus from EmbiTech (San Diego, CA) and illuminated using a blue LED from IO Rodeo (Pasadena, CA). The optical filter for visualization of Gel Green stained gels was a 540 nm rapid edge filter from Omega optics (Austin, TX). Quantitative PCR was performed on a single channel open PCR machine from Chai (Santa Clara, CA).

## Methods

Two approaches were used to obtain the exact same chemical environments favorable for purification of cfDNA. Table 1 ("liquid-based method") describes the reagents used in a solution-based approach that is convenient for small sample sizes. Magnetic beads are one of the most expensive components in the process and we found they are most effective when added in amounts proportional to volume. Therefore, to minimize costs for large volume purifications, we also devised a method in which pure and highly concentrated chemical constituents are added directly to plasma (Table 2; "solid-based method").

Both methods were performed in plastic containers and not glass; glass is itself a silica surface that can bind DNA and drastically reduce yields. The first step was to combine proteinase K (formulated at 20 mg/mL in 50 mM Tris pH 8.0, 3 mM $CaCl_2$ and 50% glycerol (v/v); store at 4˚C) and plasma. Digestion buffer reagents were then added. Reagents were dissolved by stirring and the reaction was brought to 56˚C for approximately one hour. Binding reagents were then combined followed by the addition of beads. This slurry was brought to room temperature for about 5 min and then aliquoted into 50 mL conical centrifuge tubes. The tubes were placed in a magnetic separation rack. Once the beads were aggregated into a pellet, the supernatant was poured into a bio-hazard waste vessel. The beads were then washed with wash buffer #1, wash buffer #2, and 100% ethanol and dried completely. The total DNA was eluted with 15 ul of TE (10 mM Tris pH 8.0, 0.1 mM EDTA) per 1 mL of initial plasma input.

For double-sided solid phase reversible immobilization (SPRI) DNA fragment size selection [14–16], 2 volumes of total DNA were combined with one volume of DNA purification bead solution [17,18]. These were incubated for 10 min at room temperature (RT), the beads were pulled aside, and the 3 volumes of supernatant were transferred to a vessel containing 2 additional volumes of DNA purification beads. The resulting solution was incubated for 10 min, the beads were pulled down (with bound cfDNA nucleosomal fragments), and the bead pellet was washed twice with 1 mL of 70% ethanol/water (v/v), and resuspend in 1 ul per 1 mL of plasma (the yield in ng/ul is also the original quantity in plasma in ng/mL).

For quality analysis, total DNA was purified using both the UltraPrep method and the QIAamp Circulating Nucleic Acid kit from Qiagen/Thermofisher (Hilden, Germany) as instructed by the manufacturer. RNAse A (Qiagen) treatment of QIAamp total DNA was performed by adding the enzyme directly to the total DNA (in elution buffer) to a final concentration of 10 ng/ul followed by incubation at 37˚C for 30 min. The yield of total and size-fractionated DNA was measured using a Qubit fluorometer and AccuGreen[TM] reagents from Biotium. DNA gels were performed in 2% agarose with TBE buffer and stained with Gel Green dye (Biotium). The molecular size standards were the PCR marker from New England Biolabs which are 766, 500, 300, 150 and 50 bp. Alu quantitative PCR (qPCR) was performed with primers GAGGCTGAGGCAGGAGAATCG and GTCGCCCAGGCTGGAGTG [19] with One-Taq hot start (New England Biolabs) and EvaGreen[TM] dye (Biotium). The Cq values are

**Table 1. Liquid-based method for isolation of total DNA.**

| Component | Composition | Relative volumes | 10 mL plasma prep | Cumulative volume | Notes |
|---|---|---|---|---|---|
| Plasma | | | 10 mL | 10 mL | Perform in 50 mL screw cap tube |
| Proteinase K, 20 mg/mL solution | 20 mg/mL Proteinase K, 50 mM Tris pH 8.0, 3 mM CaCl2, 50% glycerol v/v | Combine at a ratio of 1 volume Proteinase K solution per 100 volumes of plasma | 100 ul | 10.1 mL | Mix prior to adding digestion buffer |
| Digestion buffer | 5 M GITC, 25% Tween 20, 50 mM Tris pH 8.0, 25 mM EDTA | Combine at a ratio of ~2 volumes of digestion buffer per 3 volumes of plasma/ Proteinase K | 6.5 mL | 16.6 mL | Heat to 56 C for ~ 1 hour |
| Binding buffer | 3.5 M GITC, 45% isopropanol, 2.5% Tween 20, 10 mM Tris pH 8.0, 1 mM EDTA | Combine at a ratio of ~2 volumes of binding buffer to 1 volume of Plasma/Prot K/Digestion buffer | 33 mL | 49.6 mL | Mix prior to adding beads |
| 400 nm Silica beads | supplied as a 2.5 mg/mL solution | Combine at a ratio of 1 volume beads to 125 volumes of digested plasma in binding buffer | 400 ul | 50.0 mL | Mix during addition of beads. Incubate 10 min at RT |
| Wash solution #1 | 3 M GITC, 30% isopropanol, 5% Tween 20, 40 mM Bis-Tris pH 6.0, 2 mM EDTA | For every 50 mL tube | 5 mL | | Perform in 5 mL tube |
| Wash solution #2 | 50 mM Tris pH 8.0, 0.5 mM EDTA, 80% EtOH v/v | For every 50 mL tube | 5 mL | | Perform in 5 mL tube |
| 100% ethanol | | For every 50 mL tube | 1 mL | | Transfer to 1.5 mL tube. Aspirate and dry at 37 C |
| TE buffer | 10 mM Tris pH 8.0, 0.1 mM EDTA | For every 1.5 mL tube, elute with 100 ul then 60 ul | 100 ul, then 60 ul | | Anticipate volume of ~150 ul of DNA |

converted into yield-of-Alu-sequences using the equation Alu yield = power(10,-0.3*Cq+6) in Microsoft Excel. This number was divided by 0.5 ng of input DNA to calculate yield-per-ng.

**Table 2. Solid-based method for isolation of total DNA.**

| Component | Amount per 100 mL of plasma | Cumulative volume | Notes |
|---|---|---|---|
| Plasma | 100 mL | 100 mL | Perform in a plastic container with the capacity to hold > 250 mL |
| Proteinase K, 20 mg/mL solution | 1 mL | 101 mL | 20 mg/mL Proteinase K in 50 mM Tris pH 8.0, 3 mM CaCl2, 50% glycerol v/v. Mix with plasma prior to adding digestion reagents |
| 1 M Tris pH 8.0 | 2.8 mL | 140 mL | Add liquid and solid ingredients directly to the plasma/proteinase K. Heat to 56˚C. Incubate for one hour at 56˚C |
| 0.5 M EDTA | 2.8 mL | | |
| Solid GITC | 33 g | | |
| Tween 20 | 14 g | | |
| GITC | 56 g | ~250 mL | Add the GITC and isopropanol directly to the digested plasma and mix to dissolve. Then add beads, mix, and dispense into 50 mL aliquots. Incubate 10 min, pull aside beads and discard supernatant |
| isopropanol | 75 mL | | |
| 400 nm Silica beads | 2.5 mL | | |
| Wash solution #1 | 25 mL of 3 M GITC, 30% isopropanol, 5% Tween 20, 40 mM Bis-Tris pH 6.0, 2 mM EDTA | 5 mL per 50 mL tube | Resuspend pellet in each tube in 5 mL Wash #1. Pool 5 x 5 mL wash volumes = 25 mL into fresh 50 mL tube. Pull aside beads and discard supernatant |
| Wash solution #2 | 25 mL of 50 mM Tris pH 8.0, 0.5 mM EDTA, 80% EtOH v/v | | Resuspend pellet in 25 mL of Wash #2. Pull aside beads and discard supernatant |
| 100% ethanol | 5 mL | | Resuspend pellet in 5 mL 100% ethanol. Transfer to 5 mL tube. Pull aside beads, discard supernant, aspirate residual solvent and dry |
| TE buffer | 10 mM Tris pH 8.0, 0.1 mM EDTA | Elute with 1000 ul, then 600 ul | Resuspend dried pellet in 1000 ul TE, pull aside beads, and transfer supernatant into fresh 1.5 mL tube. Perform second elution of beads with 600 ul of TE and pool. Expect to recover~1500 ul of eluate |

Library construction is evaluated by monitoring the attachment of adapters containing standard Illumina P5 and P7 sequences to cfDNA using a proprietary library construction technology (Ripple Biosolutions, Seattle, WA). The attachment efficiency is evaluated using PCR primers `AATGATACGGCGACCACCGAGATCTACACTCTTTCCCTACACGACGCTCTTCC-GATCT` (Illumina-specific) and `GAGGCTGAGGCAGGAGAATCG` (Alu-specific) and qPCR conditions as described above. The results are quantified using a standard curve of premade cfDNA library material. For determination of the amount of DNA recovered, this same library reference material was spiked into plasma at a concentration of 1 ng/mL prior to extraction or into total DNA at a concentration of 5 ng/mL prior to size selection. The percent recovery relative was determined by comparison to the starting material using the qPCR methods described above.

## Results

### cfDNA extraction and enrichment

The UltraPrep cfDNA purification method described here is a two-step process. The first step is extraction of total DNA from plasma with an emphasis on near-complete recovery of DNA present in the sample (Fig 1). The second step is size-based separation of nucleosomal-sized cfDNA fragments from hmwgDNA. While the extraction technique described here is superficially similar to many other DNA extraction technologies, our method has several distinctive features. First, proteinase K is often the most expensive reagent used in DNA extraction procedures. To minimize cost, the conditions of the initial digestion step were configured to maximize proteinase activity and thereby allow reduced amounts of the enzyme to be used. This is described in more detail in S1 Fig. Second, several studies have shown that while DNA readily binds to silica surfaces in a variety of chemical conditions, a substantial fraction does so irreversibly [9,10]. Here, the chemistry favors reversible association of DNA with silica beads and therefore robust recovery (~84%) of total DNA (S2 Fig). Third, the DNA purification method is completely passive and therefore mechanical devices such as vacuum pumps or centrifuges are not required. Fourth, the method is scalable, meaning the yield of extracted DNA per milliliter of input is consistent across a broad range of sample input volumes.

We developed two related approaches for DNA extraction. For smaller sample volumes, a liquid-based protocol that utilizes additions of premade buffers is outlined in Table 1. This is the method illustrated schematically in Fig 1. For larger samples, we developed a method that involves additions of concentrated materials directly to plasma in amounts that recapitulate the chemical environment most favorable for high yield recovery of DNA (Table 2). The former approach is convenient while the latter strategy minimizes cumulative sample volume and therefore the amount of somewhat costly silica beads needed to fully recover total DNA (S3 Fig). Both methods produce comparable yields of cfDNA (S1 Table).

Total DNA extracted from plasma is most often a mixture of nucleosomal fragments and hmwgDNA. These two species can be partitioned into separate fractions using double-sided SPRI [14–16]. The exact proportions of DNA and DNA purification bead solution used in this study are shown in Fig 2. The polyethylene glycol (PEG) and to a lesser extent the salt, that is present in DNA purification bead solutions [17,18], drive binding of DNA onto the surface of carboxyl-coated magnetic beads. The core principle behind double-sided size separation is that there is an inverse relationship between the concentration of PEG and the size of bound DNA fragments. In the first step a more dilute concentration of PEG favors binding of high molecular weight DNA. The supernatant is then added to additional PEG (and SPRI beads) in the second step to recover the nucleosomal cfDNA fragments. The overall recovery of DNA from this enrichment step was about 80% (S4 Fig). This generates an estimate that about 2/3 of

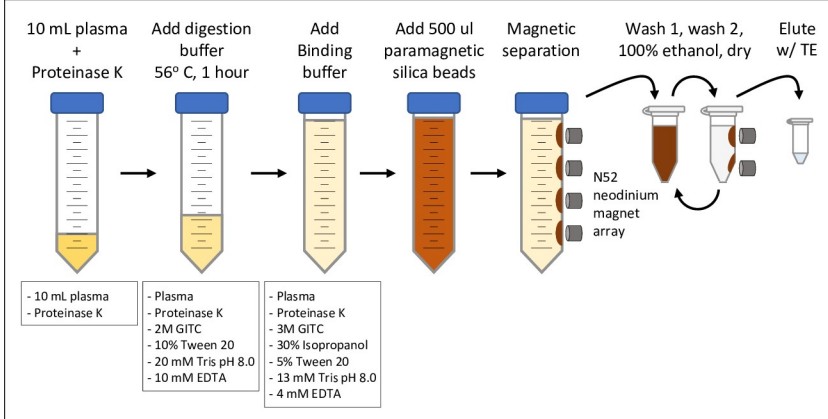

**Fig 1. UltraPrep procedure for purification of total DNA from plasma using the liquid-based method.**

the nucleosomal cfDNA fraction (84% from step1 x 80% from step 2 = 67% overall) was recovered in the UltraPrep process. Coincidentally, in the plasmapheresis samples we have worked with, about 1/3 of the total DNA is high molecular weight and 2/3 is nucleosomal cfDNA (for example, see Fig 2).

## Performance

With an eye toward both research applications and clinical utilization of the cfDNA purified using the UltraPrep method, we established four independent assays and a comparison with the industry-standard method to evaluate UltraPrep purified material (Fig 3). First, we measured the yield of double-stranded DNA (dsDNA) using dsDNA-specific fluorescent dyes and a Qubit fluorometer. The typical yield of purified, nucleosomal-sized cfDNA from healthy donor plasmas collected by automated plasmapheresis into sodium citrate was 3–4 ng per mL of plasma (S1 Table). Second, we used agarose gel electrophoresis to determine size distribution of purified material. Acceptable samples exhibited a fragmentation pattern consistent with DNAse cleavage in the linker region between adjacent nucleosomes. An example is

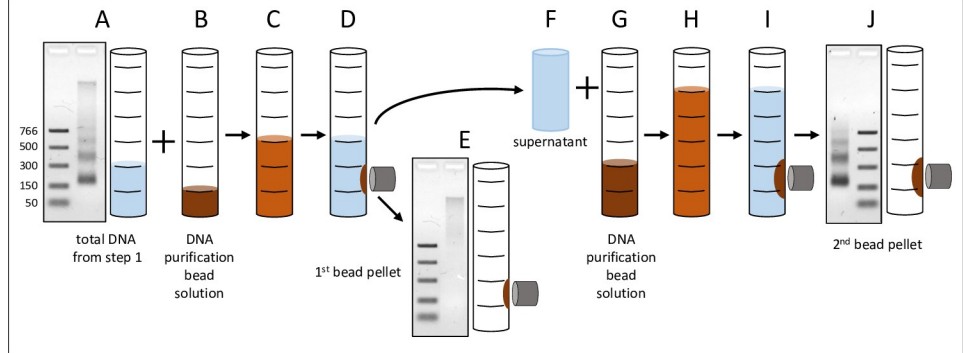

**Fig 2. Double-sided SPRI bead enrichment of nucleosomal cfDNA fragments.** (A) Two volumes of total DNA from the first stage of the UltraPrep procedure are (B) combined with one volume of DNA purification bead solution [17,18]. The numbers to the left of the gel image refer to the sizes of the molecular weight markers in bp. (C) After a 10 min incubation at RT, (D) the beads are pulled aside and the (F) three volumes of supernatant are transferred to (G) two additional volumes of DNA purification beads. (H) The blended mixture is incubated for 5 min and (I) the beads with bound cfDNA are pulled aside, washed with 70% ethanol/water, dried and (J) the DNA is eluted with TE.

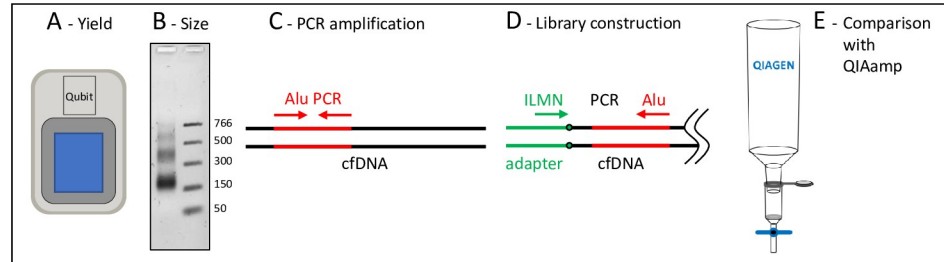

**Fig 3. Four quality assays and one comparison used to evaluate purified cfDNA.** (A) The Qubit fluorometer was used to quantify the amount of double-strand-DNA-specific dye bound to DNA. (B) Agarose gel electrophoresis was used to assess the fragmentation pattern of purified cfDNA. The numbers to the right of the gel image refer to the sizes of the molecular weight markers in bp. (C) Alu sequence-specific qPCR with primers directed to the human Alu sequence [19] were used to measure potential PCR inhibition in purified preparations of cfDNA with a readout of Alu yield detected/ng of DNA. (D) Library construction efficiency was determined by qPCR as the percentage of cfDNA ends attached to an adapter that contains standard Illumina NGS sequences. (E) An aliquot of the plasma samples used in large scale preparations was purified using the industry standard QIAamp technology, and the resulting DNA from both methods was compared using the quality assays described in (A) through (D).

shown in Fig 4. Third, several downstream analytical techniques (e.g. ddPCR, targeted amplicon sequencing, BEAMing, etc.) require that the input cfDNA is a robust amplification template devoid of inhibitors. We created an assay to detect PCR inhibition by monitoring the amplification efficiency of human Alu sequences using qPCR at a fixed input amount of purified DNA (0.5 ng/25 ul qPCR reaction). The resulting Cq values generated by the qPCR instrument are converted into yield of Alu amplicons per ng of input DNA using a simple formula (Methods). There is, of course, no "gold standard", inhibitor-free, human cfDNA sample that we could use to calibrate performance and instead we benchmarked the assay performance across several purified lots of cfDNA; the typical values in units of Alu yield/ng cfDNA were approximately 100 ± 20 (see, for example, Fig 4). Fourth, several quantitative NGS methods for cfDNA analysis rely on the attachment of adapter sequences as a prerequisite for creating genomic cfDNA libraries. The percentage of cfDNA ends that become ligated an adapter is often referred to as the "conversion rate", and high conversion rates are critical to the success of these methods. We measured percent conversion efficiency using qPCR with a primer pair where one primer was specific for standard Illumina NGS sequences present in the adapter sequence and the other was specific for the human Alu repeat. Using standard curve analysis with a fully adapted cfDNA library control, the assay measured the amount of adapter-modified cfDNA ends per total input amount of cfDNA ends. The ratio between these two values was expressed as the percent conversion efficiency. A proprietary adapter ligation method was used for these measurements (Methods). Typical values across multiple cfDNA preparations was approximately 40 ± 5% (e.g. Fig 4). Finally, most published studies cite the QIAamp Circulating Nucleic Acid kit from Qiagen as the method used for initial purification of cfDNA; in other words, this is the established purification technology by which other methods should be benchmarked. We routinely compared the assay performance metrics for cfDNA purified from the same plasma using the QIAamp procedure and the UltraPrep method.

The comparisons with QIAamp purified cfDNA merit further consideration (Fig 4). Our initial observation was that QIAamp-purified cfDNA had less specific activity for Alu content and lower rates of adapter attachment than cfDNA purified by the UltraPrep method described here. As a specific example of a direct evaluation of methods using the same plasma sample (19359), the Alu yield/ng for QIAamp material was 64 ± 11 versus 107 ± 17 for UltraPrep material (eight replicate determinations for each sample). Similarly, the adapter attachment conversion efficiencies were 19 ± 2% for QIAamp prepared cfDNA versus 39 ± 3% for

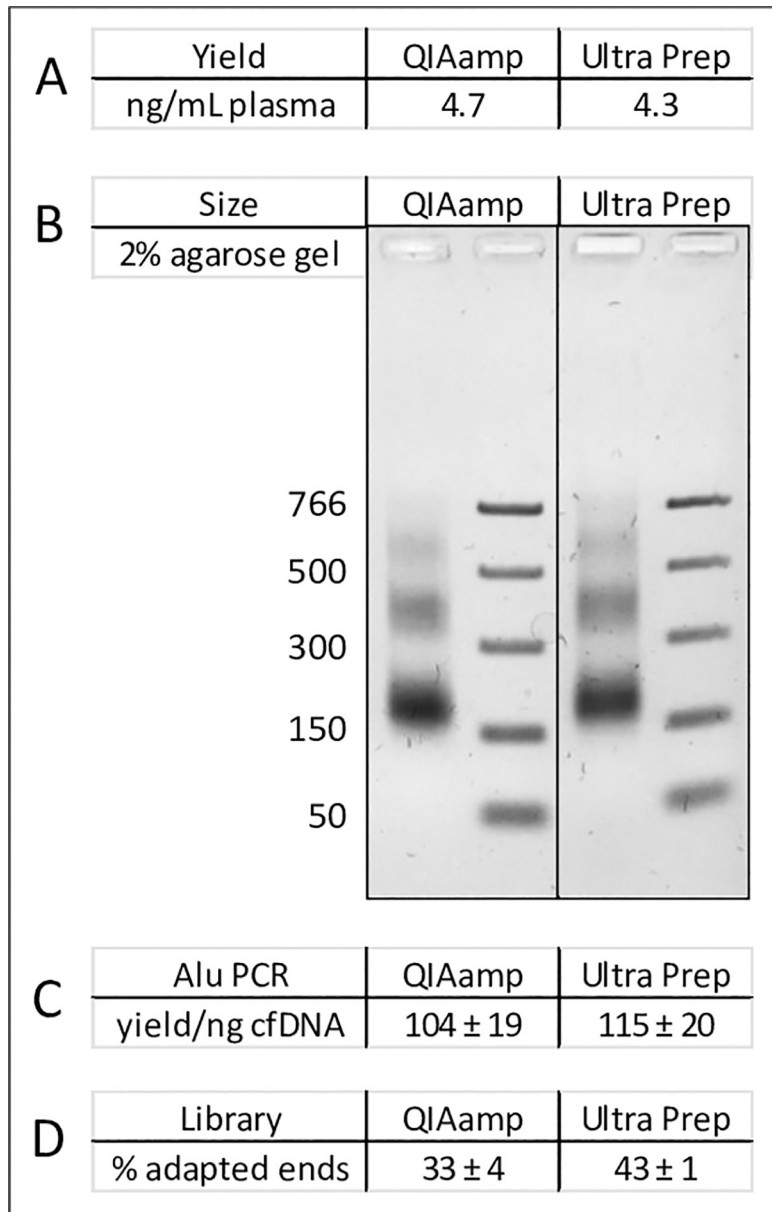

**Fig 4. Quality analysis of nucleosomal sized DNA purified from a QIAamp prep and from UltraPrep total DNA.** (A) Total yield of cfDNA fragments. The values for each sample were determined by a single measurement using a Qubit fluorometer. (B) Size distribution of the purified cfDNA. The sizes of the fragments in PCR marker standard are indicated in units of bp. (C) Comparison of Alu yields per ng of cfDNA. (D) Comparison of percent adapter attachment conversion efficiencies.

UltraPrep prepared cfDNA (three replicate measurements of each sample). Further investigation revealed two reasons for this. First, the QIAamp kit is a total DNA isolation method that collects both hmwgDNA and cfDNA fragments. High molecular weight DNA, present to some extent in many plasma samples, performs poorly in the library construction assay, thereby accounting for some of the discrepancy. Second, we found that the carrier RNA routinely added during QIAamp purification is a significant interfering substance. It falsely elevates Qubit readings, resulting in overestimation of DNA concentrations by as much as 50%.

In the example shown in Fig 4, the initial total DNA Qubit reading for the QIAamp extracted sample indicated a yield of 12.2 ng/mL plasma. After RNAse A treatment (see Methods), this value dropped to 8.3 ng/mL plasma. Using the same lot of plasma, the total DNA yield from the UltraPrep protocol that does not use carrier RNA was a comparable value of 9.3 ng/mL plasma. After size selection, the yield of nucleosomal fragments from the two methods was essentially the same (Fig 4A). Similarly, the size distribution, Alu PCR and library construction results were more or less identical for both sets of samples.

The UltraPrep method was also successfully applied to whole blood samples collected in lavender-top K$_2$EDTA vacutainer tubes (S1 Table). The yield of nucleosomal-sized cfDNA was rather high in these samples, which, based on equivalent yields from QIAamp and on previous studies [13], we believe to be a characteristic of the sample and not the collection method. The method was also applied to unspun urine that was collected in EDTA-containing vessels. Most of the resulting DNA was high molecular weight, with a broad smear present in the low molecular size fraction (data not shown).

## Discussion

The UltraPrep open-source method for purification of cfDNA represents a significant advance in the ability to access this vital diagnostic analyte. It represents a very significant reduction in cost from currently used methods. The cost of cfDNA isolation from human plasma using the current industry standard QiaAmp technology is approximately five dollars per mL of plasma processed. The total cost of reagents and consumables using the UltraPrep process is approximately 50 cents per mL of plasma processed. The yield (ng per mL plasma) of purified nucleosomal fragments from the two methods is indistinguishable. The UltraPrep protocol scales easily from a few mL of plasma to hundreds of mL of plasma with little change in the time and effort required for cfDNA purification. Small- and large-scale purifications can easily be completed in a single day. The resulting purified material performs exceptionally well in downstream analytical assays. The size selection step addresses a major sample collection preanalytical concern by substantially reducing the amount of hmwgDNA that may be present. This is significant since excess hmwgDNA can cause a significant underestimation of the minor allele frequency of rare tumor markers. In our view, size selection is preferable to using fixative-containing blood collection tubes to stabilize blood cells. The same reagents that mitigate cell breakage can potentially cross-link DNA and thereby confound test results.

The UltraPrep method makes purification of microgram quantities of cfDNA from single-donor plasmapheresis collections feasible. This in turn opens new opportunities. For instance, the same cfDNA sample can conceivably be used for diagnostic research, assay development, and testing implementation. A panel of donor samples can be used time and again to calibrate the background noise in newly developed genomic assays. This is particularly important in the case of next-generation sequencing applications where systematic error can generate false positive signals. Moreover, there is an acute need for "truth samples", comprised of *bona fide* cfDNA, that can be used for proficiency testing. The current paradigm of comparing cancer patient cfDNA with matched DNA extracted from tumor biopsies invariably generates discrepancies that are most often explained away as biological phenomenon [20]. Similarly, "synthetic cfDNA" spiked with known markers is an uncertain approximation of genuine, physiologically generated DNA [13]. Rather we propose that proficiency testing can feasibly be accomplished by monitoring common genetic polymorphisms in systematically blended cfDNA samples from two unrelated donors [5]. Lastly, our overarching goal is to see tests for early detection of cancer that are conducted during routine wellness exams. Most asymptomatic individuals are capable of donating the quantities of whole blood that will be needed for

deep genomic coverage detection tests. UltraPrep technology has the scale to accommodate these higher volume plasma samples.

## Supporting information

**S1 Fig. Yield of total DNA as a function of proteinase K addition.** Identical 10 mL aliquots of several different donor samples were processed using concentrations of proteinase K shown. The quantity of 200 ug/mL plasma (10 ul of a standard 20 mg/mL solution of enzyme added per mL of plasma) that was chosen for the protocol is highlighted in green. This amount of enzyme corresponds to four cents per mL of processed plasma, which is less than 10% of the overall cost. While there were significant yields of DNA in the no-added-enzyme control samples, these samples were "sticky" and extremely difficult to process.
(PPTX)

**S2 Fig. Recovery of spiked-in DNA from four replicates of UltraPrep.** The spike-in DNA was a completed cfDNA library with Illumina adapter sequences. This DNA can be specifically detected in a qPCR reaction by using an Illumina P5-specific primer coupled with a human Alu primer (see Methods). One ng per mL of plasma was added. The amount of library in the preprocessed control and four replicates was determined by qPCR using the Illumina + Alu primer pair. The average recovery from the four samples was 84%.
(PPTX)

**S3 Fig. Yield of total DNA as a function of silica bead concentration.** Identical 10 mL aliquots of plasma were processed using the microliters of beads/prep volume shown. The yield of total DNA after the initial purification step and of high molecular weight versus nucleosomal-sized DNA is after the size selection step are shown as a function of added bead volume (2.5 mg/mL beads). The prep volume per mL of input plasma is larger for the liquid-based prep than for the solid-based prep by a factor of two-fold. In consideration of this, the quantity of 8 ul/mL prep volume was chosen for the liquid-based prep and 10 ul/mL prep volume for the solid-based prep.
(PPTX)

**S4 Fig. Recovery of spiked-in DNA after size fractionation.** The spike-in material was completed cfDNA library with Illumina adapter sequences. Five ng per mL of total DNA was added. The amount of material eluted from the first and second bead pellets was determined by qPCR using the Illumina + Alu primer pair. "Total" recovery is the sum of the two elutions. Percentages were determined by comparison to the starting material.
(PPTX)

**S1 Table. Yields of cfDNA across time and samples.**
(XLSX)

**S1 Raw images.**
(PDF)

## Author Contributions

**Conceptualization:** Christopher K. Raymond, Kay Hill.

**Funding acquisition:** Christopher K. Raymond.

**Investigation:** Christopher K. Raymond, Fenella C. Raymond.

**Methodology:** Christopher K. Raymond, Fenella C. Raymond.

**Resources:** Kay Hill.

**Supervision:** Christopher K. Raymond, Fenella C. Raymond.

**Validation:** Kay Hill.

**Writing – original draft:** Christopher K. Raymond.

**Writing – review & editing:** Christopher K. Raymond, Fenella C. Raymond, Kay Hill.

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
