## [Decision Letter · Decision Letter 0]

28 Apr 2020

PONE-D-20-09370

UltraPrep is a scalable, cost-effective, bead-based method for purifying cell-free DNA

PLOS ONE

Dear Dr. Raymond,

Thank you for submitting your manuscript to PLOS ONE. After careful consideration, we feel that it has merit but does not fully meet PLOS ONE’s publication criteria as it currently stands. Therefore, we invite you to submit a revised version of the manuscript that addresses the points raised during the review process.

We would appreciate receiving your revised manuscript by Jun 12 2020 11:59PM. To enhance the reproducibility of your results, we recommend that if applicable you deposit your laboratory protocols in protocols.io, where a protocol can be assigned its own identifier (DOI) such that it can be cited independently in the future. For instructions see: http://journals.plos.org/plosone/s/submission-guidelines#loc-laboratory-protocols

We look forward to receiving your revised manuscript.

Kind regards,

Ruslan Kalendar, PhD

Academic Editor

PLOS ONE

Journal Requirements:

'The authors that contributed to this study are affiliated with either Ripple Biosolutions (CR and FR) or PlasmaLab International (KH) as employees. In addition, the authors have an ownership stake in these entities. This affiliation does not alter our adherence to PLOS ONE policies on sharing data and materials.'

We note that one or more of the authors are employed by a commercial company: Ripple Biosolutions and PlasmaLab International

Reviewers' comments:

Reviewer's Responses to Questions

**Comments to the Author**

1. Is the manuscript technically sound, and do the data support the conclusions?

Reviewer #1: No

Reviewer #2: Partly

Reviewer #3: Yes

Reviewer #4: No

2. Has the statistical analysis been performed appropriately and rigorously? 

Reviewer #1: No

Reviewer #2: N/A

Reviewer #3: N/A

Reviewer #4: No

3. Have the authors made all data underlying the findings in their manuscript fully available?

Reviewer #1: No

Reviewer #2: Yes

Reviewer #3: Yes

Reviewer #4: No

4. Is the manuscript presented in an intelligible fashion and written in standard English?

Reviewer #1: Yes

Reviewer #2: Yes

Reviewer #3: Yes

Reviewer #4: Yes

5. Review Comments to the Author

Reviewer #1:

Title of Reviewed Manuscript: UltraPrep is a scalable, cost-effective, bead-based method for purifying cell-free DNA

Detection of cell free DNA (cfDNA) in plasma currently termed “liquid biopsy” is a growing field with high importance in cancer diagnostics as well as prenatal diagnostics. The author’s approach is to use magnetic beads to separate the full amount of nucleic acid in the sample, then size select for the smaller freely circulating DNA by binding the longer DNA and using the remaining liquid with shorter fragments to bind with new beads by changing the binding conditions. This type of size selection has been demonstrated previously and is not a new innovation to the field. There is no discussion of other enrichment approaches previously reported in the literature and how this approach is different or adds value to the field.

The novelty of this method is the use of dry reagents added directly to plasma for large volume extraction of cfDNA. However, I would not call this approach innovative, it could be useful to those interested in extracting large volumes of plasma.

There are many concerns throughout the text with verb tense and tone, and lack of detail needed for other scientists to replicate the work. Because of this, the recommendation is to require major revisions to the manuscript.

Below are specific concerns of note:

• The size separation approach has been demonstrated before in several forms and should be properly cited by the author. The dry method is interesting and perhaps useful for larger volumes of plasma. This aspect of the study is not mentioned in the abstract.

• The author states that there is a “largely unmet need for large, well-characterized, single-donor lots of normal human cfDNA that can be used for diagnostic test research, assay development, and routine proficiency qualification in clinical laboratory environments.” However, it’s not clear how pre-size selected cFDNA would be helpful in these applications, unless this extraction method is used as the upfront processing for these downstream applications.

• Materials section:

“This seller offers an array of magnetic racks for various tube sizes that are, in our experience, the best available, and they are at a remarkably modest price point.” – Commentary is inappropriate for publication.

• Methods section:

o “Digestion reagents are then added.” Only proteinase K is mentioned in previous sentence, are there more digestion reagents?

o Paragraph reads as an SOP, but wrong tense for manuscript: “For size selection, combine 2 volumes of total DNA with one volume of DNA purification beads [13].

Incubate for 10 min at RTo C, pull aside beads, and transfer the 3 volumes of supernatant to a vessel containing 2 additional volumes of DNA purification beads. Incubate for 10 min, pull down beads (with

bound cfDNA nucleosomal fragments), wash the bead pellet twice with an appropriate volume of 70%

ethanol/water (v/v), and resuspend in 1 ul per 1 mL of plasma (the yield in ng/ul is also the original

quantity in plasma in ng/mL).”

• It is assumed that 100% high molecular weight DNA is removed, but only analyzed by gel electrophoresis, which has much worse sensitivity as compared to PCR. There could be some remaining and contributing to the PCR amplification signals that is just not visible by gel electrophoresis. This is a major concern.

• No indication of number of replicates, error bars in any of the figures.

• All of the actual data is in supplemental figure/tables. Some of these should be in the main text, especially S1 Table. However, S1 table needs more replicates and more rigorous statistical analysis.

• There is PEG mentioned in the text, but not in any of the method details or Tables. Unclear when and how this method was used in the study.

• The reviewer was unable to find these measurements of “amplifiability” and “clonability”reported later in the text.

“We used qPCR measurement of human Alu sequences to establish that the specific “amplifiability” per picogram of purified cfDNA was consistent between purified lots of material and similar to cfDNA purified using the industry standard (QIAamp). Fourth, quantitative NGS methods for cfDNA analysis rely on the attachment of adapter sequences as a prerequisite for creating genomic cfDNA libraries. We monitored the “clonability” of purified cfDNA by measuring the attachment efficiency of adapters containing standard Illumina sequences to cfDNA.”

Reviewer #2:

The paper describes a new open-source, two-step method for purification of cell-free DNA. There is a high demand in non-expensive efficient alternative approaches for cell-free DNA isolation for research purposes, and a publication may attract particular interest in this regard.

Given that reproducibility of the method suggested is the main value of the publication, the 'Materials' section could be enforced by more specific information about the items used (like Cat# or type of silica-coated superparamagnetic beads). For non-standardized items from unknown sellers (like pochekailov), quality alternatives from recognized sellers could be suggested.

Plasma separation and storage conditions may play a key role in quality cfDNA extraction, so authors are suggested to add this information about conditions used in their experiments.

Although some comparison with the leader on a market has been performed, the manuscript will certainly benefit if authors are following one of the formal standardized frameworks for validation of the molecular tests (like https://www.ncbi.nlm.nih.gov/pmc/articles/PMC3002854/, but more could be available).

I am not aware of Research Ethics procedures in commercial organizations and cannot comment on this topic.

Reviewer #3:

Raymond and co-workers authors describe a method for purification of cfDNA that includes selective enrichment of smaller fragments thought to derive from cleavage of human DNA between nucleosomes of apoptotic cells.

Two protocols have been described, for smaller and larger input plasma volumes. The authors compared the performance of the UltraPrep metrics for cfDNA purified from the same plasma using the QIAamp procedure. The cfDNA obtained by the two methods generated similar quality metrics but with 10 times lower cost for UltraPrep.

Here below are my comments:

Page 2

Another potential domain of application of cfDNA is detection of bacterial pathogens (bacteraemia, sepsis). This may be mentioned in the introduction.

“… preanalytical collection conditions often result in plasma that is a mixture of high molecular weight genomic DNA and cfDNA”. This sentence is slightly unclear.

“This seller offers an array of magnetic racks for various tube sizes that are, in our experience, the best available, and they are at a remarkably modest price point. “ I would remove this sentence (looks like an advertisement).

Page 5, Table 1

Please check if 400 nM (nanomolar) Silica beads is correct (also in Table 2). Or you meant 400-nm diameter?

Is proteinase K at 1/50 plasma volume? Or 1/100?

Silica beads are at 1/100 or 1/125 volume?

Remove “.” in “TE Buffer”/“Notes” cell; add the degree symbol where required; if you capitalise the first letter of “isopropanol” than apply the same for “glycerol”.

Page 6, Table 2

The text should be revised.

In “Plasma”/”Notes” delete “cf DNA purification”.

“Digestion step”/“Notes” may be clarified. For instance, If I correctly understood: “Mix the 4 components and heat in a hot water bath at XXX °C to dissolve GITC and Tween 20. Cool to 56 °C, mix with plasma/proteinase K and incubate at 56 °C for 1 hour. (“Monitor temperature and do not exceed 56 C.” may be deleted). Clarify if the components are mixed together prior to adding to plasma/proteinase K or they are added sequentially. If mixed, merge the 4 cells “Cumulative volume” for the 4 digestion reagents (140 mL).

In “TE Buffer”/”Notes” add “eluate” after “DNA”.

grams - g

Do 50-mL-tubes accommodate 50 mL + 5 mL (Wash 1) + 5 mL (Wash 2) + 1 mL (ethanol) volume?

Page 9, Caption Fig 4.

“Comparison of Alu units per pg “. “Please indicate what is measured in pg (e.g. ”pg of XXX”).

Page 10, Discussion

In addition to cfDNA purification, plasmapheresis and downstream sequencing or ddPCR generate costs. What percentage of the overall cost is due to cfDNA purification with Qiagen and UltraPrep approaches?

What does UltraPure correspond to?

Figure 2

The figure would be easier to read if you indicate the content of tubes A, B and G directly in the figure.

Typos

“comnparison” on page 8

RT° C - room temperature (RT)

Use mL (or ml) but not both

mLs - mL

In tables and figures, add space before “M“ (molar) or “mM”

Reviewer #4:

The authors attempt to describe a novel method for purifying small sized nucleic acids from blood and plasma samples. While this is an interesting proposition, the manuscript is not written in a manner consistent with a scientific level publication. Namely, key data from experiments simply is not presented. As such I am unable to fully assess its scientific merits. I want to underscore that any and all data need to presented - key data should be in the manuscript itself and any additional data in the supplemental material. I have highlighted this, along with other points below. One note, please check the supplemental material files - my Powerpoint told me the files are corrupted.

Summary:

RESULTS

1. “First, proteinase K is often the most expensive reagent used in DNA extraction procedures”

a. S1 Fig shows that the lowest concentration was already giving the best results. Why not try lower doses?

b. Others have reported that Proteinase K may not even be needed: http://pubmed.gov/28923054. The main difference here may be the use of a final conc of 2% Triton. If cost is really a concern, it may be worth doing a quick experiment to see if Prot K can be left out. This may have the tangential benefit of preserving the small nuc acids which may degrade during Prot K treatment and reducing the length of the protocol.

2. “Fourth, the method is scalable, meaning the yield of extracted DNA per milliliter of input is consistent across a broad range of sample input volumes.”

a. Was this shown experimentally?

3. “Coincidentally, in the plasmapheresis samples we have worked with, about 1/3 of the total DNA is high molecular weight and 2/3 is nucleosomal cfDNA (for example, see Fig 2).”

a. Show the data that backs up this statement

4. Fig 3:

a. Where’s the data?

5. “First, the QIAamp kit is a total DNA isolation method that collects both high molecular weight genomic DNA and cfDNA fragments”

a. This data should be shown.

6. “Second, we found that the carrier RNA routinely added during QIAamp purification is a significant interfering substance”

a. The Qiagen handbook states that “carrier RNA present in the extracted nucleic acids is likely to dominate UV absorbance readings”, so this is already known. But this shouldn’t be a factor because as stated in the methods section, eluted samples are treated with RNAse and the dye used for Qubit is “dsDNA-specific”. Please explain.

7. Performance section:

a. Please present the data

METHODS

1. Two additional Supplemental files should be provided:

a. 1) A Word document that outlines each protocol in detail (ie step by step)

b. 2) A Excel table that lists the reagents and equipment used for the protocols (along with corresponding catalogue #)

2. A suitable replacement from a commercial vendor needs to be given for the item from Ebay as labs in some countries may not be able to procure from Ebay

3. Please explain in more detail regarding your spike-in libraries. I could not fully understand how you carried out these experiments and how you made the comparisons for percent recovery.

OTHER:

1. “comnparison” – typo

2. Show DNA ladder sizes on your figures

3. Show proper statistics, including standard deviations (I’m assuming experiments were done in replicates and at least 2 independent times)

4. What are you staining you gels with? Is it a ds-DNA specific stain or does it also stain RNA?

6. PLOS authors have the option to publish the peer review history of their article (what does this mean?). If published, this will include your full peer review and any attached files.

Reviewer #1: No

Reviewer #2: No

Reviewer #3: No

Reviewer #4: No

---

## [Decision Letter · Decision Letter 1]

14 May 2020

PONE-D-20-09370R1

UltraPrep is a scalable, cost-effective, bead-based method for purifying cell-free DNA

PLOS ONE

Dear Dr. Raymond,

Thank you for submitting your manuscript to PLOS ONE. After careful consideration, we feel that it has merit but does not fully meet PLOS ONE’s publication criteria as it currently stands. Therefore, we invite you to submit a revised version of the manuscript that addresses the points raised during the review process.

Authors need to prepare responses to the comments of reviewer # 4, including his previous comments. And prepare the manuscript in accordance with these comments.

We would appreciate receiving your revised manuscript by Jun 28 2020 11:59PM. To enhance the reproducibility of your results, we recommend that if applicable you deposit your laboratory protocols in protocols.io, where a protocol can be assigned its own identifier (DOI) such that it can be cited independently in the future. For instructions see: http://journals.plos.org/plosone/s/submission-guidelines#loc-laboratory-protocols

We look forward to receiving your revised manuscript.

Kind regards,

Ruslan Kalendar, PhD

Academic Editor

PLOS ONE

Reviewers' comments:

**Comments to the Author**

Reviewer #4: (No Response)

2. Is the manuscript technically sound, and do the data support the conclusions?

Reviewer #4: Partly

3. Has the statistical analysis been performed appropriately and rigorously? 

Reviewer #4: No

4. Have the authors made all data underlying the findings in their manuscript fully available?

Reviewer #4: No

5. Is the manuscript presented in an intelligible fashion and written in standard English?

Reviewer #4: Yes

6. Review Comments to the Author

Reviewer #4:

Authors have not sufficiently addressed my concerns. I cannot in good faith recommend this article for publication. (The authors should know that even though the files were deemed corrupted by my Powerpoint, I was able to open them as google docs and was able to assess their quality.)

---

## [Author Response · Author response to Decision Letter 1]

15 May 2020

Editor Kalendar, We provided a cover letter (the same file was attached as "response to reviewers") describing our response. The second set of comments from Reviewer #4 were not specific and therefore not actionable. As such, we did not revise the revision #1 manuscript in any way. Please help us comply with the requirements of the journal (I included a cell phone #). We believe this paper has merit and will generate a favorable response from your readership. All the best, Chris Raymond

---

## [Editor Report · Decision Letter 2]

18 May 2020

UltraPrep is a scalable, cost-effective, bead-based method for purifying cell-free DNA

PONE-D-20-09370R2

Dear Dr. Raymond,

We are pleased to inform you that your manuscript has been judged scientifically suitable for publication and will be formally accepted for publication once it complies with all outstanding technical requirements.

With kind regards,

Ruslan Kalendar, PhD

Academic Editor

PLOS ONE

---

## [Editor Report · Acceptance letter]

21 May 2020

PONE-D-20-09370R2 

UltraPrep is a scalable, cost-effective, bead-based method for purifying cell-free DNA 

Dear Dr. Raymond:

I am pleased to inform you that your manuscript has been deemed suitable for publication in PLOS ONE. Congratulations! Your manuscript is now with our production department. 

With kind regards,

on behalf of

Dr. Ruslan Kalendar 

Academic Editor

PLOS ONE